

# Detection of fungicide resistance to fludioxonil and tebuconazole in *Fusarium pseudograminearum*, the causal agent of Fusarium crown rot in wheat

Na Zhang[1], Yiying Xu[1,2], Qi Zhang[1], Le Zhao[1], Yanan Zhu[1], Yanhui Wu[1], Zhen Li[1] and Wenxiang Yang[1]

[1] College of Plant Protection, Hebei Agricultrual University, Baoding, Hebei, China
[2] Shangqiu Institute of Technology, Shangqiu, Henan, China

## ABSTRACT

*Fusarium* crown rot (FCR) on wheat is a soil-borne disease that affects the yield and quality of the produce. In 2020, 297 *Fusarium pseudograminearum* isolates were isolated from diseased FCR wheat samples from eight regional areas across Hebei Province in China. Baseline sensitivity of *F. pseudograminearum* to fludioxonil ($0.0613 \pm 0.0347$ μg/mL) and tebuconazole ($0.2328 \pm 0.0840$ μg/mL) were constructed based on the *in vitro* tests of 71 and 83 isolates, respectively. The resistance index analysis showed no resistance isolate to fludioxonil but two low-resistance isolates to tebuconazole in 2020. There was an increased frequency of resistant isolates from 2021 to 2022 based on the baseline sensitivity for tebuconazole. There was no cross-resistance between fludioxonil and tebuconazole. This study provides a significant theoretical and practical basis for monitoring the resistance of *F. pseudograminearum* to fungicides, especially the control of FCR.

## INTRODUCTION

*Fusarium* crown rot (FCR), a soil-borne disease, causes major yield losses in wheat (*Triticum aestivum* L.) worldwide. The occurrence of FCR has been reported in many arid and semi-arid wheat-growing continents and countries, including Australia (*Khudhair et al., 2021*), America (*Smiley et al., 2005*), Africa (*Gargouri et al., 2011*), Europe (*Agustí-Brisach et al., 2018*), the Middle East (*Hameed, Rana & Ali, 2012*; *Gebremariam et al., 2018*), and China (*Li et al., 2012*; *Zhang et al., 2015a*; *Xu et al., 2018*). The infected wheat results in brown necrosis at the first two or three inter-nodes and produces blighted white heads and abortive seeds when severe (*Scherm et al., 2013*). Consequently, significant yield losses occur.

Specifically, Hebei Province, among the wheat-corn rotation region in the Huang-Huai Plain, China, accounts for about 10% and 11.4% of the planting area and production in China, respectively. Cases of FCR have been reported across all the eight main wheat

Corresponding author
Wenxiang Yang,
wenxiangyang2003@163.com

regional areas in Hebei Province, leading to potential yield loss. *Fusarium* spp. including *F. pseudograminearum*, *F. colmorum*, *F. graminearum*, etc. (*Agustí-Brisach et al., 2018*; *Kazan & Gardiner, 2018*; *Scherm et al., 2013*; *Zhang et al., 2015a*) causes FCR. According to *Deng et al. (2020)*, *F. pseudograminearum* was the most isolated pathogen causing crown rot of wheat with strong pathogenicity in China. This species has been spreading in Hebei Province and has been repeatedly shown to be associated with *Fusarium* head blight (FHB) as well (*Xu et al., 2015*, *2018*; *Ji et al., 2016*).

Seed dressing chemicals for controlling FCR diseases in China, such as Qingxiu (10% difenoconazole), Cruiser (2.2% fludioxonil + 2.2% difenoconazole), Dividend (3% difenoconazole), Raxil (6% tebuconazole), Celest (2.5% fludioxonil), and Aobairui (1.1% tebuconazole) are used in controlling FCR disease in China. The most commonly used active ingredients are tebuconazole, difenoconazole, and fludioxonil. As for the characteristics of these chemicals, fludioxonil belongs to the phenylpyrrole class of chemistry and has a unique mode of action by inhibiting the phosphorylation of glucose, resulting in the inhibition of the growth of fungal mycelium. Its use has also been shown to increase the seed emergence rate of wheat (*Hysing & Wiik, 2014*). This chemical has been commercialized in China since 2013.

Triazole fungicides, tebuconazole and difenoconazole for instance, are fungicides characterized by high efficiency, wide spectrum, safety, long duration, and strong internal absorption. They are sterol 14α- demethylase inhibitors (DMIs), which affect ergosterol biosynthesis. Recently, some reports showed that DMIs were the most effective chemical for controlling diseases caused by *Fusarium* spp. (*Delen, 2016*; *Hellin et al., 2017*), and can also be used to prevent the formation of mycotoxins produced by *F. culmorum* and *F. graminearum* (*Shah et al., 2018*). Fungicides containing DMIs have been used in the USA, Europe, and China for many years. Some registered commercial products with tebuconazole as the active ingredient include Raxil (6% tebuconazole), Liangshi (1.1% tebuconazole and 19.9% imidacloprid), Aobairui (1.1% tebuconazole and 30.8% imidacloprid). These commercial agents have been used for many years to control diseases such as sharp eyespot, *Fusarium* head blight, and powdery mildew. It has also been registered as seed dressing for FCR control.

Fungal pathogens may develop resistance to different fungicides under specific selection pressures or under conditions of adversity (*Feng et al., 2020*). Resistance to fludioxonil has been reported in a broad range of plant pathogenic fungi such as *Colletotrichum gloeosporioides* from fruit (*Schnabel et al., 2021*), *Sclerotinia sclerotiorum* from oilseed rape (*Kuang et al., 2011*), *Botrytis cinerea* from apple and strawberry (*Zhao et al., 2010*; *Fernández-Ortuño et al., 2016*). A range of DMI-resistant fungal strains have been reported from pathogenic populations of *Botrytis cinerea* (*Zhang et al., 2020*), *Pseudocercospora fijiensis* (*Chong et al., 2021*), *F. graminearum* (*de Chaves et al., 2022*), *Monilinia fructicola* (*Lesniak et al., 2020*), and *Venturia nashicola* (*Ishii et al., 2021*).

Currently, chemical control of FCR is the most effective method to limit disease. Still, repeated fungicidal applications may reduce the sensitivity to *Fusarium* isolates to the fungicides and thus increase the risk of severe plant disease. The determination of the susceptibility of pathogenic *Fusarium* species to fungicides in wheat has focused on

**Table 1 Sensitivity to fludioxonil for 71 *F. pseudograminearum* isolates from eight different geographical regions across Hebei Province in China.**

| Geographic regions | Isolates no. | EC$_{50}$ lowest | EC$_{50}$ highest | Ratio of highest to lowest | Mean value |
|---|---|---|---|---|---|
| Xingtai (XT) | 7 | 0.0369 | 0.0870 | 2.36 | 0.0617 ± 0.0177[c] |
| Cangzhou (CZ) | 10 | 0.0261 | 0.0872 | 3.34 | 0.0503 ± 0.0205[b] |
| Baoding (BD) | 6 | 0.0281 | 0.0474 | 1.69 | 0.0399 ± 0.0078[a] |
| Tangshan (TS) | 6 | 0.0319 | 0.1651 | 5.18 | 0.0687 ± 0.0496[d] |
| Handan (HD) | 7 | 0.0362 | 0.1789 | 4.94 | 0.0754 ± 0.0479[e] |
| Hengshui (HS) | 7 | 0.0423 | 0.0981 | 2.32 | 0.0606 ± 0.0188[c] |
| Langfang (LF) | 7 | 0.0322 | 0.1009 | 3.13 | 0.0605 ± 0.0282[c] |
| Shijiazhuang (SJZ) | 21 | 0.0165 | 0.1719 | 10.42 | 0.0688 ± 0.0439[d] |

**Note:** Different lowercase letters marked following the mean values identify significantly different means (Duncans'new multiple-range test, $p < 0.05$, $n = 3$).

*F. graminearum*, the cause of *Fusarium* head blight (FHB) (*de Chaves et al., 2022*; *Breunig & Chilvers, 2021*). In China, *Yin et al. (2021)* showed that carbendazim strongly inhibited *F. pseudograminerum* populations, with a baseline sensitivity of 0.755 ± 0.336 μg/mL. However, little information is available about the activity and the risk of resistance to fludioxonil and tebuconazole in *F. pseudograminearum*. Therefore, this study aimed to evaluate such sensitivity and cross-resistance for *F. pseudograminearum* field populations to fludioxonil and tebuconazole and monitor the resistance of *F. pseudograminearum* isolates to tebuconazole. Results from this research may provide the first reference for the resistance monitoring of the pathogen, as well as the rational application of these fungicides for controlling wheat crown rot worldwide, especially across different regions within Hebei Province in China.

## MATERIALS AND METHODS

### Collection of *F. pseudograminearum* isolates

From late April (heading stage) to late May 2020 (filling stage), when wheat stems exhibited characteristic crown rot symptoms, diseased samples were collected from different wheat-grown regions including Xingtai, Cangzhou, Baoding, Tangshan, Handan, Hengshui, Langfang, and Shijiazhuang across Hebei Province in China (Table 1). The infected stalks were sampled randomly for an individual isolate, with at least 30 isolates obtained from each region and a minimum geographical distance of at least 2 km between any two sample sites. A total of 297 field isolates were isolated according to the method described by *Deng et al. (2020)*. The single-spore isolate was obtained and cultured on the PDA medium for each isolate. Species identifications of 272 strains (accounting for 91.6%) were confirmed as *F. pseudograminearum* using primers Fp1-1 and Fp1-2 (*Demeke et al., 2005*) and the amplicon sequence analysis of *EF1* and *EF2* (*Proctor et al., 2009*).

### Preparation of fungicide-containing medium

Technical grade fludioxonil (98% active ingredient (a.i.)) and tebuconazole (97% a.i.) were used for the *in vitro* sensitivity assay. Stock solutions of fludioxonil were obtained by dissolving the original chemical with methyl alcohol to 1,000 mg/mL. Tebuconazole was

dissolved with acetone to obtain the same concentration. PDA plates were amended with fludioxonil to give serially final concentrations of 0.015, 0.03, 0.06, 0.12, 0.24, and 0.48 μg a.i./mL. Other PDA plates was amended with tebuconazole with concentrations of 0.025, 0.1, 0.4, 1.6, and 6.4 μg a.i./mL, while control PDA plates were amended with 0.1% (v/v) methyl alcohol or acetone (*Liu, Hai & Jiang, 2016*) only.

## Baseline sensitivity of *F. pseudograminearum* to fludioxonil and tebuconazole

For the sensitivity test, at least six isolates from each wheat geographic region were randomly selected to form a subset population. Seventy-one *F. pseudograminearum* isolates were tested against fludioxonil and 83 isolates against tebuconazole using the mycelial growth rate method described by *Secor & Rivera (2012)*. Generally, 0.7 cm mycelial plugs from the edge of actively growing fungal colonies were transferred upside down onto the center of PDA plates amended with fludioxonil or tebuconazole. The diameters of the colonies were measured for each treatment by criss-cross after 3–4 days of incubation at 27 °C in the dark. The *in vitro* experimental design was completely randomized consisting of three replications for each treatment and was repeated twice. The effective concentration for 50% growth inhibition ($EC_{50}$) was calculated using the fungicide concentrations and the corresponding inhibition rate of mycelial growth. Colony diameter (cm) = measured colony diameter-fungal plug diameter (0.7 cm). Relative inhibition (%) = [(colony diameter of control − colony diameter of treatment)/colony diameter of control] × 100. Fungicide concentrations (μg/mL), converted into a base-10 logarithmic value ($x$). The inhibition of mycelial growth was analyzed by the Statistical Package of the Social Science (SPSS21.0) software to make a linear regression of the corresponding probability value of the colony growth inhibition percentage against the $Log_{10}$-transformed fungicide concentration (*Liu, Hai & Jiang, 2016*). The final baseline sensitivity was established using the average $EC_{50}$ values of the isolates, which fit the normal distribution (*Hu et al., 2020*).

## Fungicides resistance isolates and their frequency

The fungicides resistance index for each isolate was assessed by the formula below. The resistance of *F. pseudogramineum* to fludioxonil and tebuconazole can be divided according to the following criteria, and samples classified as low resistance (LR), midium resistance (MR) and high resistance (HR) were all taken as fungicide resistance isolates. Resistance index (RI) = $EC_{50}$ of the tested isolate/baseline sensitivity (*Li et al., 2021*). Sensitive isolate (S): 0 < RI ≤ 5, LR isolate: 5 < RI ≤ 10, MR isolate: 10 < RI ≤ 40, HR isolate: 40 < RI. Frequency of resistant isolates (%) = (resistant isolates/total number of tested isolates) × 100.

## Cross-resistance analysis

A subset of 65 *F. pseudograminearum* isolates was used to assess their cross-resistance. The linear regression analysis was carried out using $lgEC_{50}$ of fludioxonil to the strain as the X-axis and $lgEC_{50}$ of tebuconazole to the strain as the Y-axis, and the linear regression
equation y = bx + a was constructed. For the determination of the Pearson coefficient (r) and the significance level of the independent sample T-test (*P* value), cross-resistance between fludioxonil and tebuconazole were analyzed.

## Monitoring of resistance isolates

Ten sensitive isolates in the year 2020 were selected for sensitivity assay by measuring minimum inhibitory concentration (MIC) (*Taga et al., 1982*). MIC was estimated by observing mycelial growth three days after inoculation on the medium amended with tebuconazole concentrations of 0, 1.0, 5.0, 10.0, 15.0, 20.0, and 25.0 µg/mL. When all 10 isolates were completely inhibited, the corresponding concentration was further tested for 107 isolates randomly selected from the field population in 2021 (49 isolates) and 2022 (58 isolates).

## Data analysis

The SPSS21.0 and Microsoft Office Excel 2010 programs package were used for statistical analysis. The means of results were calculated for no significant difference ($p < 0.05$) observed in mycelial growth for the two experiments. Pearson correlation analysis was carried out using the SPSS21.0 software, and Duncan's new complex range method was used to test the significance of differences.

# RESULTS

## Sensitivity of mycelial growth for *F. pseudograminearum* to fludioxonil

The $EC_{50}$ values for all 71 isolates were combined to establish a sensitivity baseline. The $EC_{50}$ values of the corresponding isolates for mycelial growth assays were continuous, ranging from 0.0165 to 0.1789 µg/mL, with a mean value of 0.0613 ± 0.0346 µg/mL. The variation factor (the ratio of the maximum to the minimum $EC_{50}$ values) was 10.84. Based on the $EC_{50}$ value of the tested isolates, the frequency distribution showed a unimodal curve (Fig. 1). The isolates with $EC_{50}$ values in the range of 0.03–0.06 µg/mL showed the highest frequency (54.93%). The average $EC_{50}$ value of 0.0613 µg/mL was preliminarily determined as the baseline sensitivity of *F. pseudograminearum* to fludioxonil. No resistant isolate of *F. pseudograminearum* was observed in the field subset population.

The mean $EC_{50}$ values of *F. pseudograminearum* isolates collected from different geographic regions were significantly different (Table 1). The isolates with the most sensitivity (<0.03 µg/mL) were from Shijiazhuang, Baoding, and Cangzhou within Hebei Province in China. The isolates with the highest $EC_{50}$ were from Shijiazhuang and Handan. Isolates from Baoding showed the lowest sensitivity variation to fludioxonil, while isolates from Shijiazhuang showed the highest.

## Sensitivity of mycelial growth for *F. pseudograminearum* to tebuconazole

The $EC_{50}$ values of 83 isolates for mycelial growth assays to tebuconazole were also continuous, ranging from 0.0417 to 1.5072 µg/mL. The variation factor was 50.21. Based

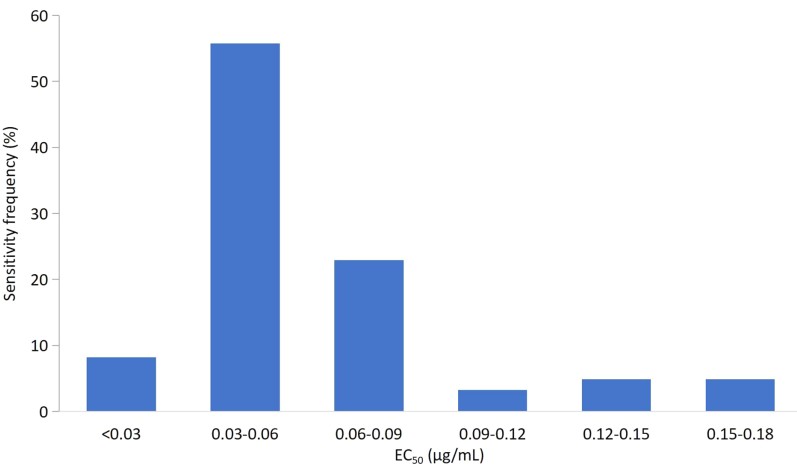

**Figure 1 Frequency distribution of the effective concentration for 50% growth inhibition (EC$_{50}$) for fludioxonil on 71 _F. pseudograminearum_ isolates.**

on the EC$_{50}$ values of the tested isolates, the frequency showed an abnormal distribution (Fig. 2) and was confirmed by SPSS21.0 ($k$ = 0.002, $p$ < 0.5). Fifty-five isolates with EC$_{50}$ values in the range of 0.04–0.40 µg/mL showed the highest frequency (66.27%). With further analysis of the frequency distribution of these 55 isolate, a unimodal curve with a positive skew was constructed (Fig. 3). The average EC$_{50}$ value of 0.2328 µg/mL for this subset of 55 _F. pseudograminearum_ isolates was preliminarily determined as the baseline sensitivity for tebuconazole with _F. pseudograminearum_. Isolates from Hengshui showed the lowest sensitivity variation on tebuconazole, while isolates from Tangshan presented the highest record (Table 2).

## Resistance index and cross-resistance analysis

The resistance index (RI) was analyzed based on the constructed sensitivity baselines of the two fungicides. Our results showed that the RI of all the 71 strains to fludioxonil was lower than 5, ranging from 0.269 to 2.918, indicating that all these strains were sensitive to fludioxonil. The RI values for 81 strains ranged from 0.179 to 4.672, indicating their sensitivity to the fungicide tebuconazole. Specifically, the two isolates (accounting for 2.41%) with the RI values of 6.196 and 6.474, these two isolates with low resistance (LR) were collected from Shijiazhuang (SJZ9) and Tangshan (TS70), respectively.

From the isolates we tested, a subset of 65 isolates was used for cross-resistance analysis using the SPSS21.0. The result showed that there was no correlation ($r$ = 0.295), at a significant difference ($p$ < 0.05), between fludioxonil and tebuconazole (Fig. 4). This result also means that there was no cross-resistance between these two chemical agents tested.

## Resistance isolates from the field population

From the ten sensitive isolates, 3 (20HS16, 20TS65, and 20CZ237) were completely inhibited at 5.0 µg/mL, 3 (20SJZ39, 20HD77, 20LF223) at 10.0 µg/mL, 2 (20HD14 and 20SJZ208) at 15.0 µg/mL and 20BD38 at 20.0 µg/mL. A total of 25.0 µg/mL was confirmed as the minimum inhibitory concentration (MIC), considering that all 10 isolates were

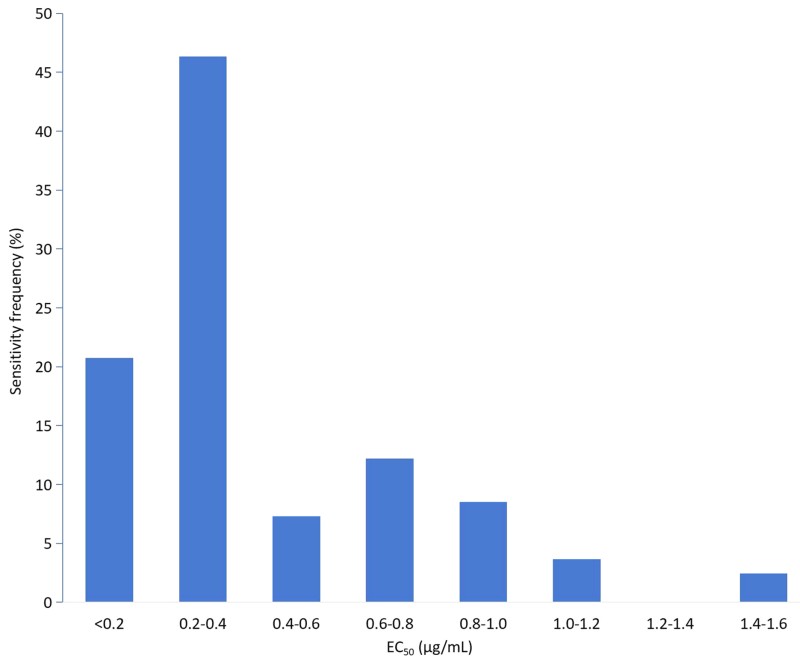

**Figure 2 Frequency distribution of the effective concentration for 50% growth inhibition (EC$_{50}$) for tebuconazole on 83 _F. pseudograminearum_ isolates.**

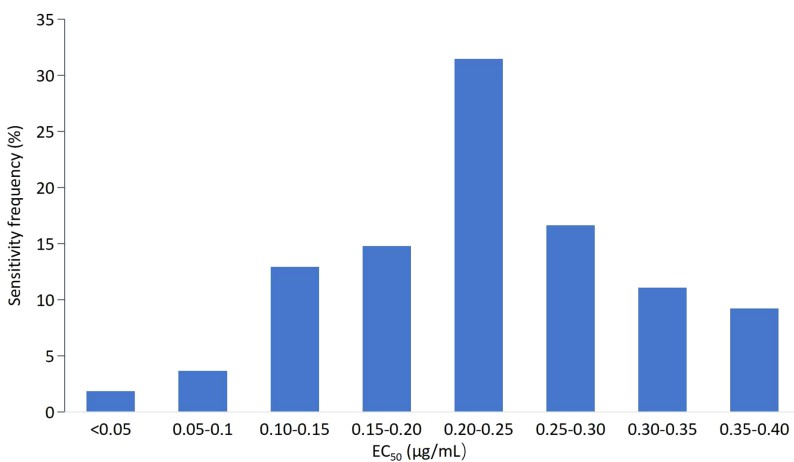

**Figure 3 Frequency distribution of the effective concentration for 50% growth inhibition (EC$_{50}$) for tebuconazole on a subset of 55 (B) _F. pseudograminearum_ isolates.**

completely inhibited at this concentration. When the 107 isolates were tested, totally 22 isolates (11 out of 49 in 2021, and 11 out of 58 in 2022) survived on plate amended with tebuconazole (25 μg/mL). These 22 isolates were then further tested _in vitro_ assay on mycelial growth. Based on the resistance index, seven low resistance isolates (accounting for 6.54%) were resistance isolates (Table 3). In detail, two isolates (accounting for 4.08%) from Shijiazhuang (SJZ) and Xingtai (XT) in 2021, and five (8.62%) from Hengshui (HS), Shijiazhuang (SJZ), Handan (HD), and Cangzhou (CZ) in 2022 were detected (Table 4).

**Table 2 Sensitivity to tebuconazole for 83 *F. pseudograminearum* isolates from eight different geographical regions across Hebei Province in China.**

| Geographic regions | Isolates no. | EC$_{50}$ lowest (μg/mL) | EC$_{50}$ highest (μg/mL) | Ratio of highest to lowest | Mean value |
|---|---|---|---|---|---|
| Xingtai (XT) | 7 | 0.1130 | 0.7810 | 6.91 | 0.4431 ± 0.3048[e] |
| Cangzhou (CZ) | 10 | 0.1476 | 1.0876 | 7.37 | 0.4756 ± 0.3607[g] |
| Baoding (BD) | 6 | 0.1388 | 0.3344 | 2.41 | 0.2677 ± 0.0729[b] |
| Tangshan (TS) | 7 | 0.0417 | 1.5072 | 36.13 | 0.4590 ± 0.5064[f] |
| Handan (HD) | 8 | 0.2341 | 0.5497 | 2.35 | 0.3785 ± 0.1344[d] |
| Hengshui (HS) | 7 | 0.0990 | 0.4985 | 5.03 | 0.2293 ± 0.1479[a] |
| Langfang (LF) | 7 | 0.2198 | 0.3849 | 1.75 | 0.2958 ± 0.0622[c] |
| Shijiazhuang (SJZ) | 31 | 0.0682 | 1.0533 | 15.43 | 0.5169 ± 0.3562[h] |

Note:
Different lowercase letters marked following the mean values identify significantly different means (Duncans' new multiple-range test, $p < 0.05$, $n = 3$).

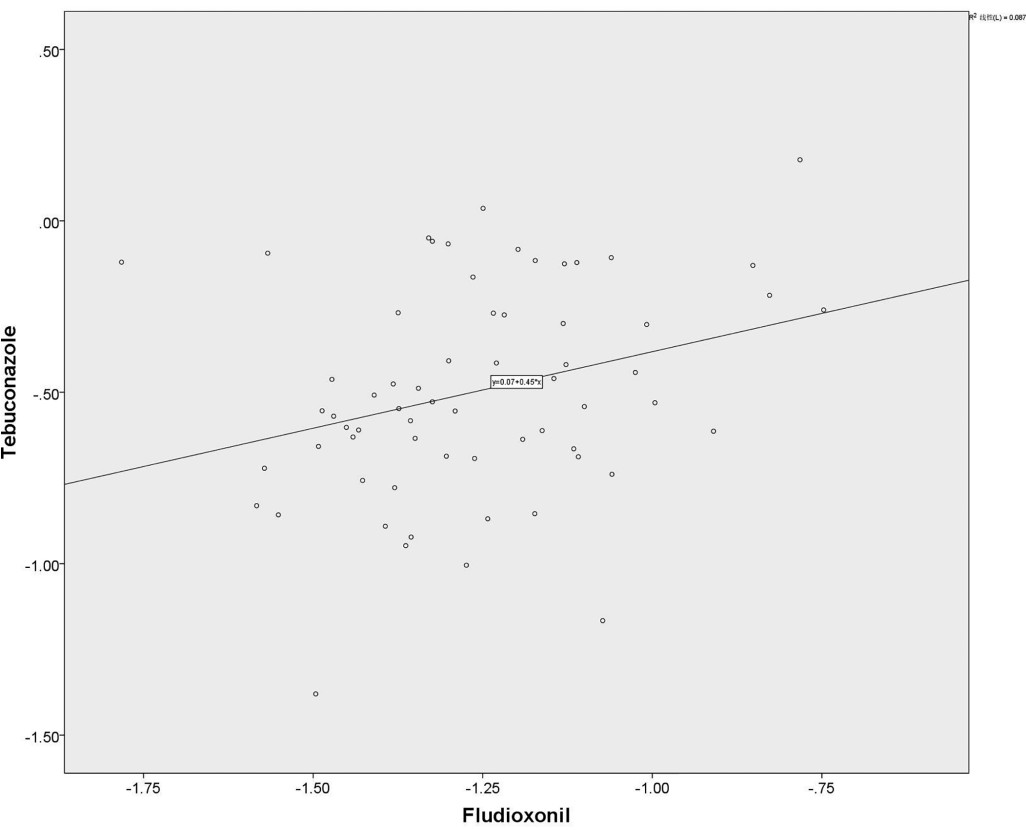

**Figure 4 Result from the cross resistance analysis between fludioxonil and tebuconazole.**

## DISCUSSION

Baseline sensitivity data of a phytopathogenic fungus to a fungicide are useful for evaluating the risk of resistance developing in sensitive populations of the fungus (*Zhang et al., 2015b*). In this study, the EC$_{50}$ values for fludioxonil to *F. pseudograminearum*

Table 3 Fields monitoring of sensitivity of *Fusarium pseudograminearum* isolates to tebuconazole.

| Isolates | EC$_{50}$ (µg/mL) | Resistance index[a] | Isolates | EC$_{50}$ (µg/mL) | Resistance index |
|---|---|---|---|---|---|
| 21HS140 | 0.9199 | 3.95 | 22HS18 | 1.3205 | 5.67[b] |
| 21HS328 | 0.8641 | 3.71 | 22HD2 | 1.8443 | 7.92[b] |
| 21HD77 | 1.0596 | 4.55 | 22HD4 | 1.1155 | 4.79 |
| 21HD80 | 1.0179 | 4.37 | 22HD8 | 1.4465 | 6.21[b] |
| 21CZ356 | 0.8825 | 3.79 | 21CZ605 | 1.1117 | 4.78 |
| 21CZ367 | 0.9778 | 4.20 | 21CZ608 | 1.7939 | 7.71[b] |
| 21SJZ120 | 1.1899 | 5.11[b] | 21CZ609 | 1.1371 | 4.88 |
| 21BD117 | 0.8038 | 3.45 | 21CZ620 | 1.1585 | 4.98 |
| 21BD202 | 1.1295 | 4.85 | 22SJZ460 | 1.1557 | 4.96 |
| 21XT63 | 1.1551 | 4.96 | 22SJZ462 | 1.2804 | 5.50[b] |
| 21XT96 | 1.3107 | 5.62[b] | 22SJZ463 | 1.1078 | 4.76 |

Notes:
[a] Resistance index derived from EC$_{50}$ of the isolate divided by 0.2328 (the baseline sensitivity).
[b] *Fusarium pseudograminearum* isolates showed low resistance to tebuconazole.

Table 4 Monitoring of *Fusarium pseudograminearum* recovered from wheat fields in Hebei.

| Geographical region | Number of isolates | 2020 | | 2021 | | 2022 | |
|---|---|---|---|---|---|---|---|
| | | S[a] | LR | S | LR | S | LR |
| SJZ | 46 | 29 | 1 | 7 | 1 | 7 | 1 |
| HD | 24 | 8 | 0 | 8 | 0 | 6 | 2 |
| HS | 23 | 7 | 0 | 8 | 0 | 7 | 1 |
| CZ | 25 | 10 | 0 | 7 | 0 | 7 | 1 |
| XT | 23 | 7 | 0 | 7 | 1 | 8 | 0 |
| BD | 24 | 6 | 0 | 10 | 0 | 8 | 0 |
| TS | 16 | 5 | 1 | /[b] | / | 10 | 0 |
| LF | 7 | 7 | 0 | / | / | / | / |
| Total | 188 | 79 | 2 | 47 | 2 | 53 | 5 |

Notes:
[a] S and LR indicate sensitive, and low resistance to tebuconazole.
[b] indicates no samples were collected.

ranged from 0.0165 to 0.1789 µg/mL. Such differences may be related to the natural differences of the strains in different regions, the physiological differences in the isolates themselves, as well as the population structure of the *F. pseudograminearum* isolates under control level in each wheat production region (*Feng et al., 2020*).

In this research, the variation factor between the most sensitive and the least sensitive isolate was 10.42, indicating that the species *F. pseudograminearum* was sensitive to fludioxonil in nature. Since the baseline sensitivity results in this study was 0.0613 µg/mL, and there was no fludioxonil resistant *F. pseudograminearum* isolate detected, such result could be used for monitoring any future sensitivity shifts in resistance to fludioxonil in the field populations of *F. pseudograminearum*. Meanwhile, it provides further evidence to indicate effective fungicides and future methods for controlling of fungicide-resistant mutants. Another significant finding from this research showed a low resistance (LR)

frequency (2.41%) on *F. pseudograminearum* to tebuconazole from the field population in 2020 and an increasing frequency of low resistance in 2021 (4.08%) and 2022 (8.62%). This result indicated that rotational and substitution strategies for fungicides with other modes of action should be implemented to delay the development of serious resistance.

There are various ways to reduce the use of fungicides in controlling FCR. One of the primary methods is to provide detailed information, including active ingredients, potential targets, and risk exposures for different types of pesticides used for seed treatments (*Lamichhane & Laudinot, 2021*). Generally, clarifying the cross-resistance of a pathogen to different fungicides will also help provide a theoretical basis for prolonging fungicides used to control pathogens (*Feng et al., 2020*). Based on our results, there is no cross-resistance between fludioxonil and tebuconazole. The natural population of *F. pseudograminearum* in Hebei Province was most sensitive to fludioxonil *in vitro*. By contrast, the high variation factor of tebuconazole (50.21) suggests that there may be different levels of control of wheat disease within different wheat production regions. In the meantime, low resistance isolates from the field population to tebuconazole suggest that further consideration should be given to prohibiting tebuconazole as the active ingredient in wheat seed dressings. Our former research also indicated that Raxil and Dividend (tebuconazole and difenoconazole as the active ingredient, respectively) showed relatively lower control efficacy compared with Celest (2.5% fludioxonil) under a pot assay (*Zhang et al., 2022*). Applying fludioxonil in mixtures with newer fungicides, other than triazole fungicides, such as pydiflumetofen or even biocontrol agents, may reduce the risk of developing fungicide resistance in *F. pseudograminearum*.

## CONCLUSIONS

This is the first report on the baseline sensitivity of *F. pseudograminearum* populations to fludioxonil and tebuconazole from China. Fungicides with fludioxonil have been used successfully to control wheat crown rot in recent years. No cross-resistance for these two agents with *F. pseudograminearum* was recorded. The baseline sensitivity (0.0613 μg/mL for fludioxonil established in this study can be used to detect the further resistance level for field populations. Based on the baseline sensitivity of tebuconazole (0.2328 μg/mL), a total of 4.76% low resistance isolates were monitored from year 2020–2022, which guides our rational use of the appropriate fungicides.

## ACKNOWLEDGEMENTS

The authors are grateful to Shuming Luo and Percy Wong (University of Sydney) for the critical review of this manuscript.

### Funding

The study was funded by Modern Agricultural Industry System of Wheat Industry in Hebei Province Innovation Team (HBCT2018010204) and Key Research and

Development Project Hebei Province (19226507D). The funders had no role in study design, data collection and analysis, decision to publish, or preparation of the manuscript.

### Grant Disclosures
The following grant information was disclosed by the authors:
Modern Agricultural Industry System of Wheat Industry in Hebei Province Innovation Team: HBCT2018010204.
Key Research and Development Project Hebei Province: 19226507D.

### Competing Interests
The authors declare that they have no competing interests.

### Author Contributions

- Na Zhang conceived and designed the experiments, authored or reviewed drafts of the article, and approved the final draft.
- Yiying Xu performed the experiments, prepared figures and/or tables, and approved the final draft.
- Qi Zhang performed the experiments, prepared figures and/or tables, and approved the final draft.
- Le Zhao performed the experiments, prepared figures and/or tables, and approved the final draft.
- Yanan Zhu analyzed the data, authored or reviewed drafts of the article, and approved the final draft.
- Yanhui Wu analyzed the data, prepared figures and/or tables, and approved the final draft.
- Zhen Li analyzed the data, prepared figures and/or tables, and approved the final draft.
- Wenxiang Yang conceived and designed the experiments, authored or reviewed drafts of the article, and approved the final draft.

### Data Availability
The raw measurements are available in the Supplemental Files.

### Supplemental Information
Supplemental information for this article can be found online at http://dx.doi.org/10.7717/peerj.14705#supplemental-information.

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
