# Peer review of "Detection of fungicide resistance to fludioxonil and tebuconazole in Fusarium pseudograminearum, the causal agent of Fusarium crown rot in wheat"

_PeerJ, doi:10.7717/peerj.14705_

## Round 0.1 · original submission · Major Revisions

Dear Dr. Zhang:

Thank you for your submission to PeerJ. Three experts had reviewed your manuscript and reviews came back in splits. Both reviewers 1 and 2 had recommended MINOR and MAJOR revision, respectively but reviewer 3 had denied publication. I appreciate the hard work in generating and conducting research. It is my opinion as the Academic Editor for your article - Exploration of fungicide resistance to Fludioxonil and Tebuconazole of Fusarium pseudograminearum, the causal agent of wheat Fusarium crown rot - that it requires several Major Revisions.

My suggested changes (see the attached pdf file) and reviewers' comments are shown below and, on your article, on the 'Overview' screen. In addition, extensive English editing is required.
Thank you for considering PeerJ for the publication of your research.

Best regards,

Tika Adhikari

·

Basic reporting

1. I consider that the contribution is pertinent, only the grammar should be checked.

2. In line 46, check if the common name of the fungicide is Dividend instead of Dividan since there is no record of a fungicide with that name.

3. The corrections that must be made to the contribution are minor.

Experimental design

I consider that the statistical analyses are adequate.

Validity of the findings

No comment

Reviewer 2 ·

Basic reporting

This paper by Zhang et al. describes the collection of Fusarium Pseudograminearum isolates from Hebei province, China, that cause Fusarium crown rot (FCR) in Wheat, randomly selected the isolates from each location and did in-vitro plate assay to discover the baseline sensitivity (EC50) of the fungus to commonly used fungicides Fludioxonila and Tebuconazole to discover if any resistance isolates are developing in the region to these compounds. The author reported a baseline sensitivity of 0.061 µg/mL for Fludixonil and 0.2328 µg/mL for Tebuonazole based on an in-vitro test of 61 and 82 isolates, respectively. No resistance isolates to Fludioxonil were found however 2 isolates with very low resistance were detected for Tebuconazole based on resistance index analysis. No cross-resistance between these two fungicides was detected by evaluating 54 isolates. The author claimed that the finding of this study would help monitor the resistance of this fungus to fungicides and control of FCR.

General comments:
First, writing needs to be improved a lot, focusing mostly on grammar, structure, and spelling too. Some sentences are vague, and spellings are incorrect throughout the manuscript. Please re-write the manuscript.

The introduction missed a lot of literature review and background information to show the study's rationale. A literature review is not sufficient. Please rewrite it.

Figures and tables are not stand alone. The numbering in the figure is incorrect in a few cases. The table and figure don't provide sufficient information as it should.

Experimental design

The experimental design is not explained clearly. How many replication, which design? What is the parameter for the fungal incubation during the in-vitro assay etc.? Please rewrite.

Validity of the findings

The results and conclusion do not match the materials and method. It needed to be reorganized and written coherently. Please re-write.

Additional comments

General comments:
First, writing needs to be redone, focusing mostly on grammar, structure, and spelling too. Some sentences are vague, and spellings are incorrect throughout the manuscript.

FCR causal organisms in the first paragraph. Isn’t it a complex disease caused by multiple other pathogens? Doesn’t it infect other cereals as well? Please review and rewrite. (Kazan and Gardiner, 2018. https://bsppjournals.onlinelibrary.wiley.com/doi/epdf/10.1111/mpp.12639).

Materials and Methods:
The author reported that random sampling was performed, and at least 30 isolates/locations (8 locations) with at least 10 km apart. How many diseased plant samples were collected per location? Please provide a wheat cultivar name and what was the crop stage during collection. Expand table 1 with more information about the isolates, isolates id, and any pertinent information.

Why was Carbendazim not included in the study, which has a strong inhibitory effect on F. pseudograminearum reported in other parts of China?


Please explain how these 297 isolates collected were confirmed as F. pseudograminearum. Did the author use morphological or molecular-based ITS sequencing for identification?

Please explain the reason for 2x dilution for Fludioxonil and 4x dilution for Tebuconazole. Please provide any prior studies or references.

Baseline Sensitivity test:
Please explain why the different numbers of isolates (61 for Fludioxonil and 82 for Tebuconazole) were selected for the baseline sensitivity test assay for these two fungicides.

Please mention the research design and replications, and repetition.

Line 95: Baseline sensitivity was established using the frequency distribution of EC50 values.

Line 100. Please abbreviate LR, MR, and HR.

Fungicide resistance index
Please provide references for this calculation (formula below).
Formula: RI=EC50 of the tested isolate/baseline sensitivity
Please provide the data for the fungicide resistance index.

Cross-resistance analysis
Please explain why only 54 isolates were used. What is the basis of selection for these small subsets?

Results:
The author only reported the data for baseline sensitivity. Please put the results from the fungicide resistance index and cross-resistance analysis.
Line 140. Please correct Figure 2. Under figure 2, you wrote Fig 1B.

The cross-resistance analysis is not correct.

Line 148: Does a p-value greater than 0.01 means significantly different?

Annotated reviews are not available for download in order to protect the identity of reviewers who chose to remain anonymous.

Reviewer 3 ·

Basic reporting

Paper needs clear writing.
Needs additional year of isolate collection for comparison as they reported low level of resistance to Tebuconazole.
Not enough data for publication fit.

Experimental design

Looks good. Three reps and two independent experiments were conducted. However, it is not clear how they analyzed data for two independent experiments.

Validity of the findings

Need additional year of field monitoring.

Additional comments

none

Annotated reviews are not available for download in order to protect the identity of reviewers who chose to remain anonymous.

---

## Round 0.2 · Minor Revisions

Thank you for waiting and for your patience. It took longer than expected. I would appreciate it if you could check the font size and figure cation/levels and
references carefully.

Thank you.

Reviewer 3 ·

Basic reporting

ok

Experimental design

ok

Validity of the findings

ok

Additional comments

I've attached a pdf file for some edits.
Please increase the font size of the figure levels and reduce the space between bars.
Please check the reference carefully.
Thanks

Annotated reviews are not available for download in order to protect the identity of reviewers who chose to remain anonymous.

---

## Round 0.3 · accepted · Accept

Dear Dr. Zhang,

Thank you for your submission to PeerJ.

I am pleased to inform you that your manuscript - Detection of fungicide resistance to fludioxonil and tebuconazole in Fusarium pseudograminearum, the causal agent of Fusarium crown rot in wheat - has been accepted for publication in PeerJ.

Thank you again for your patience and understanding.

Congratulations !!!